# 1–2 Drinks Per Day Affect Lipoprotein Composition after 3 Weeks—Results from a Cross-Over Pilot Intervention Trial in Healthy Adults Using Nuclear Magnetic Resonance-Measured Lipoproteins and Apolipoproteins

**DOI:** 10.3390/nu14235043

**Published:** 2022-11-27

**Authors:** Trine Levring Wilkens, Zabrina Ziegler, Violetta Aru, Bekzod Khakimov, Snædís Lilja Overgaard, Søren Balling Engelsen, Lars Ove Dragsted

**Affiliations:** 1Department of Nutrition, Exercise and Sports, Section of Clinical and Preventive Nutrition, Faculty of Science, University of Copenhagen, Rolighedsvej 26, 1958 Frederiksberg C, Denmark; 2Department of Food Science, Faculty of Science, University of Copenhagen, Rolighedsvej 26, 1958 Frederiksberg C, Denmark

**Keywords:** HDL, apoA1, apoB, moderate alcohol consumption, cardiovascular disease, NMR

## Abstract

Alcohol consumption ranging from 1–2 drinks/day associates with a lower risk of coronary heart disease in some studies. The underlying mechanisms are unclear. The Metabolic Imprints of Alcoholic Beverages (MetAl) trial aimed to explore the short-term effects of moderate alcohol consumption on cardiovascular biomarkers. A 2 × 3-week cross-over single-blinded intervention trial investigating the effect of 1–2 drinks/day (~12–24 g) compared with abstention on ^1^H Nuclear Magnetic Resonance-measured main lipoproteins and subfractions was performed in 26 healthy adults. Volunteers were classified as occasional or habitual drinkers based on their habitual alcohol intakes (<2 or ≥2 drinks/week). Compared with abstention, 1–2 drinks/day increased HDL_2a_-C (*p* = 0.004), HDL_3_-C (*p* = 0.008), and HDL non-significantly (*p* = 0.19). Total apoA1 and apoA1 in HDL and its subfractions increased (*p* < 0.05). Novel findings were a decreased apoB/apoA1 ratio (*p* = 0.02), and increased HDL_2a_ phospholipid content (*p* = 0.04). In women alone, the results were similar but attenuated, and LDL-P decreased. Thus, changes in apoA1- and HDL-related biomarkers occur within weeks in moderate drinkers. Compared with abstention, 1–2 drinks/day increased total apoA1 more strongly than HDL-C and increased the cholesterol, apoA1, and phospholipid content of several HDL subfractions. Whether this provides a cardiovascular benefit requires further study. Clinicaltrials.gov: NCT03384147.

## 1. Introduction

Moderate, regular intake of 1–2 alcoholic drinks per day compared with non-drinking is associated with ~20% lower risk for coronary heart disease (CHD) in middle-aged men and women [1,2,3]. The relationship is described with a J-curve and spans ischemic stroke [4], type 2 diabetes [5], metabolic syndrome [6], and all-cause mortality [2,7]. The lowest CHD risk appears to exist at alcohol intakes between 1–2 drinks per day (up to 30 g/day), even though higher intakes also associate with a lower risk of CHD [2,3]. A lower mortality risk is observed at up to 2 and 4 drinks per day in women and men, respectively [7]. The advised maximal level of alcohol intake is generally up to 1 drink per day in women and 2 drinks per day in men in most countries [8].

Several short-term [9] and a few longer-term human alcohol intervention studies [10] show changes in cardiovascular biomarkers, such as plasma lipoproteins. Despite such effects of alcohol on lipoproteins, this may not translate into disease prevention or explain the J-shaped curve characterizing alcohol intake and CVD risk, so these relationships remain to be fully elucidated [9,11]. Lipoproteins are usually considered central in the primary and secondary prevention of cardiovascular disease (CVD) and in nutritional and therapeutic therapies [12]. Human intervention studies consistently find increased levels of high-density lipoprotein (HDL) cholesterol concentration (-C) and its major structural protein, apolipoprotein A1 (apoA1), following alcohol consumption [9,13,14]. Furthermore, HDL-C was a planned compliance marker in the Moderate Alcohol and Cardiovascular Health (MACH15) trial [15]. HDL-C is inversely associated with CHD risk, but clinical trials targeting to raise HDL-C have failed to document any cardiovascular benefit [16].

The relevance of lipoproteins in CVD may differ according to lipoprotein composition and size [17,18]. Lipoproteins are structurally dissimilar molecules with distinct metabolic functions [19]. Interestingly, lipoprotein subfractions measured via proton (^1^H) nuclear magnetic resonance (NMR) spectroscopy [20] were recently included in risk scores predicting the incidence of type 2 diabetes [21] and cardiovascular events [22].

Several observational studies, but only a few intervention studies, have investigated the effect of alcohol consumption on lipoprotein subfractions in different populations [23,24,25,26,27,28]. Results from intervention studies identifying new plausible lipoprotein biomarkers could provide indirect, mechanistic evidence explaining the potential benefit of low-moderate drinking on cardiovascular health. It is also uncertain how soon lipoproteins and their subfractions change after a shift to moderate alcohol intake or to abstention.

We aimed to investigate the effects of 1–2 drinks of alcohol per day on cardiovascular biomarkers in the cross-over intervention trial, Metabolic Imprints of Alcoholic Beverages (MetAl). MetAl primarily aimed to investigate hormone and biomarker kinetics as well as new sampling methods following short-term changes in alcohol intake (to be published). Lipoproteins and apolipoproteins were included as secondary endpoints. In the current analysis, we investigated changes in circulating levels of lipoproteins, lipoprotein subfractions, and apolipoproteins after short-term moderate alcohol intake compared with abstention.

## 2. Materials and Methods

### 2.1. Study Design and Participants

The MetAl trial was a prospective, assessor-blinded, cross-over intervention trial. The trial was with allocated intervention order based on self-reported habitual alcohol intake. Between January and May 2016, non-alcohol naïve participants aged 20–70 years were recruited from the Copenhagen area through poster boards at educational institutions and study recruitment websites. Assessed via the Timeline Followback questionnaire, eligible subjects were classified as habitual drinkers if their habitual alcohol intake (last 12 months) were ≥2 drinks per week and no more than 1 drink per day in women and 2 drinks per day in men. Participants with a habitual alcohol intake of <2 drinks per week were allocated to the group of occasional drinkers.

MetAl was designed as a cross-over trial with no wash-out period, as the primary aim was to investigate acute changes in alcohol biomarkers when going from abstention to drinking and vice versa. The trial involved a 14-day run-in period, during which participants were instructed to maintain their habitual alcohol intake. The run-in period was planned to be four days in the protocol but changed to 14 days because here we have data to evaluate alcohol intake through the 14 –day period as a more solid basis for comparison. Figure 1 provides an overview of the study design and timeline. The MetAl trial was registered at clinicaltrials.gov (NCT03384147) prior to study enrollment.

### 2.2. Eligibility Criteria

Eligible individuals had consumed alcohol within the preceding year and were able to use a smartphone app for compliance registration. Volunteers were ineligible if they were alcohol naïve or fulfilled one or more of the following criteria: past or current chronic diseases; severe psychiatric illness; regular or frequent use of medication (except over-the-counter drugs, contraceptive pills, and selective serotonin reuptake inhibitors). Additional exclusion criteria were a history of alcohol abuse according to the Alcohol Use Disorders Identification Test; intolerance or allergy to alcoholic beverages; liver blood function tests >1.5 times the normal upper limit; a breast cancer risk score >5% at screening, assessed via the Breast Cancer Risk Assessment Tool. Pregnant and lactating women, or women planning pregnancy, were ineligible.

### 2.3. Study Procedures

Potential participants were pre-screened by telephone to assess general health, risk behavior related to alcohol, and habitual alcohol intake. The Alcohol Use Disorders Identification Test questionnaire and the short Alcohol Dependence Scale (ADS) questionnaires were administered at the screening visit. The volunteers were instructed in the study procedures, including self-collection of urine and blood. Ten days later, beverages and sampling materials were distributed to eligible subjects who initiated the last four days of the run-in period while still keeping their habitual alcohol intake.

Subjects still eligible after the 14-day run-in period attended a baseline visit, at which an extensive alcohol survey was administered, including the Yale-Brown Compulsive Scale for heavy drinking, the Timeline Followback, and the Alcohol Use Disorders Identification Test questionnaires. The survey included details concerning habitual drinking patterns, frequency of alcohol intake, and the amount consumed on each occasion. Problem behavior with alcohol intake was also assessed. The participants were asked about any adverse events related to alcohol intake at all study visits.

### 2.4. Intervention

The intervention was divided into a drinking and an abstention period, with a cross-over between the two (Figure 1). Habitual drinkers initiated their intervention with three weeks of abstention from days 0–21 (period 1) and reversed to three weeks of moderate drinking from days 22–42 (period 2). The occasional drinkers followed the reverse order. The alcohol intervention was 1 drink per day for women, and 2 for men, as women generally have lower body weight and less body water compared with men [29].

The participants consumed five different alcoholic beverage types during the drinking period, and the sequence of these five different types of beverages was randomly allocated to each individual. The order of beverage types was randomized by random number allocation using Rand() function in Excel to the study-ID numbers in advance of recruitment. The recruiting personnel was blinded to the link between study ID and beverage order, and the participants were provided with their study IDs in the order of recruitment. The codes and periodic drink allocations were kept by staff involved only in the task of providing the drinks to participants.

Each beverage type was consumed for 4–5 days, and the participant could choose among different variants of each beverage type: cider (apple or pear), spirits (vodka, gin, whiskey, or snaps), white wine (sweet or dry), red white (dry or not), and beer (regular or strong). Other beverages than those offered within each category were allowed if the required volume was consumed and the participants paid themselves for the beverages (Table 1). One drink was defined as 12 grams of pure ethanol, regardless of beverage type. The randomization of beverage type was performed to support biomarker development related to different beverages (pending results) and to support the hypothesis that the metabolic effects of moderate drinking are caused by ethanol per se and not by specific beverages.

The alcoholic beverages were provided free of charge at the study visit before the three-week drinking period, and the participants were instructed in standard volumes of different alcoholic beverages, as described in Table 1. Self-reported compliance was assessed via the Timeline Followback calendar interview with reports of drinking frequency and number of drinks on each occasion [30]. Compliance during the run-in phase, the drinking period, and the abstention period was assessed at the baseline visit and the study visits on days 21 and 42, respectively. The participants were asked to maintain all other lifestyle behaviors during the study.

### 2.5. Analysis of Lipids and Lipoproteins Using ^1^H NMR Spectroscopy

#### Quantification of Lipids and Lipoproteins Based on ^1^H NMR Spectra

Proton (^1^H) NMR spectra were measured at the Department of Food Science (University of Copenhagen) using a Bruker Avance III 600 MHz NMR spectrometer equipped with a 5 mm broadband inverse probe. The spectrometer was equipped with an automated sample changer (SampleJet^TM^, Bruker BioSpin, Ettlingen, Germany) with sample cooling (278 K) and preheating stations (298 K). A detailed description of the chemicals used and the sample preparations is available in the Appendix A. Probe cooling was controlled by the BCU05 (Bruker Cooling Unit from Bruker, Billerica, Massachusetts, U.S.), with a temperature stability of 0.01 K. Data acquisition and processing were performed in the TopSpin software (Bruker, Rheinstetten, Germany). Automation of the overall measurement procedure was controlled by iconNMR™ (Bruker BioSpin, Rheinstetten, Germany). All calibrations and experiments were completed using the Bruker in vitro diagnostics research methods [31].

Briefly, 1D ^1^H NMR spectra were measured using the pulse sequence for water suppression 1D NOESY (*noesygppr1d*, Bruker nomenclature) and 32 scans, which were collected into 131,072 data points using a spectral width of 30 ppm, a 90° pulse, and a 4 s recycle delay (d1). The receiver gain value was determined experimentally and kept constant for all samples (RG = 90.5). After Fourier Transform, automatic phasing and baseline corrections were performed for all samples, followed by exponential filtering (LB = 0.3 Hz). The reader is referred to Dona et al. (2014) for further details on the instrument calibration and NMR measurements [32]. NMR spectra were imported to the SigMa software [33], scaled to their ERETIC signal [34], and aligned towards alanine’s doublet (1.507−1.494 ppm) using icoshift [35]. Lipoprotein main fractions and subfractions were quantified according to Khakimov et al. (2021) [36].

The densities of the lipoprotein fractions were as follows: very-low-density lipoprotein (VLDL), <1.006 kg/L; intermediate-density lipoprotein (IDL), 1.006–1.019 kg/L; low-density lipoprotein (LDL), 1.019–1.063 kg/L; and HDL, 1.063–1.210 kg/L. LDL was separated into six fractions: LDL_1_, 1.019–1.031 kg/L; LDL_2_, 1.031–1.034 kg/L; LDL_3_, 1.034–1.037 kg/L, LDL_4_, 1.037–1.040 kg/L, LDL_5_, 1.040–1.044 kg/L, LDL_6_, 1.044–1.063 kg/L. HDL was separated into three subclasses: HDL_2b_, 1.063–1.100 kg/L; HDL_2a_, 1.100–1.125 kg/L; and HDL_3_, 1.125–1.210 kg/L [37]. The apolipoprotein B (apoB) molecular weight of 550 kDa was used for estimating the particle number concentration (-P) of apoB-containing lipoproteins via the formula: particle number = (apoB-containing lipoprotein mg/dL × 10,000)/550.

### 2.6. Statistical Methods

Data and models were checked for normal distribution based on histograms, residual plots, and normal probability plots. Continuous variables were presented in mean (±SD) for normally distributed variables and median (range) for non-normally distributed variables. Categorical variables were summarized using absolute numbers (%). We used non-paired *t*-tests for continuous variables with normal distribution and The Mann–Whitney U test for non-normally distributed variables in baseline analyses. Chi-squared test or Fisher’s exact test was used for categorical variables. Before and after analyses for each group in each period were analyzed with a paired t-test or Wilcoxon signed-rank test.

We used a mixed model for unbalanced, repeated measurements for testing the mean change in lipoproteins and apolipoproteins following alcohol intake. The model included a three-way period-time-treatment interaction term as a fixed effect and subjects as a random effect. *Treatment* referred to either drinking or abstention, *time* covered before or after each intervention, and *period* was defined as either period 1 (day 0 to the average of day 21 and 22), or period 2 (average of day 21 and 22 to day 42). The models were fitted with Residual Maximum Likelihood, and available case analyses were performed for all outcomes.

In the primary analysis, drinkers were compared with abstainers, independent of group relationship and period. In the secondary analyses, the intervention and abstention groups were compared in each of the two three-week periods. In sensitivity analyses, models were adjusted for confounding by sex and grams of alcohol per kg body weight. In addition, a period-subject interaction was included as a random effect. The same analyses were performed in women only, i.e., occasional drinkers were compared with habitual drinking women. Lastly, carry-over was evaluated by a t-test, comparing the between-group averages of a variable after period 1 and period 2.

For descriptive purposes, correlations among all lipids and lipoproteins were illustrated in a heatmap. All analyses were performed in R (R Development Core Team, version 4.0.0) [38] with the extension packages *multcomp* [39], *lme4* [40], *ggplot2* [41], *ggpubr* [42], and *dplyr* [43]. The significance tests were two-sided, and *p* values < 0.05 were considered statistically significant.

#### Sample Size

The power calculation was based on the primary outcome, dehydroepiandrosterone sulfate (DHEAS), which will be reported elsewhere. The analysis indicated that 24 participants would be required to detect a significant (10%) difference in DHEAS with a power of 95% and a significance level of 0.05. Taking potential 10% non-compliance and 10% drop-out into account, the recruitment goal was 30 volunteers.

## 3. Results

### 3.1. Baseline Characteristics

Between January and May 2016, 66 individuals were screened by telephone, and 29 were eligible for inclusion (Figure 2). Based on habitual alcohol intake at baseline, 12 participants were allocated to the occasional drinkers’ group and 17 to the habitual drinkers’ group. One participant was initially classified as an occasional drinker but moved to the habitual drinkers´ group before the intervention to join her partner among habitual drinkers. She was classified as a habitual drinker before the initiation of any intervention and is listed among habitual drinkers in the flowchart (Figure 2). Two occasional drinkers and one habitual drinker dropped out during the first period of the study. Thus, in the final analysis, 10 and 16 participants were included in the occasional drinkers´ and habitual drinkers´ groups, respectively (Figure 2). Table 2 shows the included participants´ baseline characteristics for occasional drinkers, habitual drinkers, and habitual female drinkers only. The occasional drinkers´ group comprised women only, resulting in significant sex and height differences between the groups. No participant reported antihypertensive or lipid-lowering medication use. There were no differences between groups in plasma lipids at baseline, but occasional female drinkers had higher BMI than female habitual drinkers.

### 3.2. Compliance and Adverse Events

Except for the three drop-outs, all women drank 1 drink per day (~12 g/day) and men 2 drinks per day (~24 g/day) during the three-week intervention period (Table 1). All volunteers kept their habitual drinking habits during the initial 10 days of the 14-day run-in period. During the last 4 days, most participants were abstinent in both groups, but the alcohol intake was significantly different between groups during the 14 days prior to baseline, *p* = 0.03. The alcohol intake among women only was not different at baseline, *p* = 0.12 (Table 2). During the three-week alcohol intervention, compliance was 100% according to the self-reports obtained via the Timeline Followback interview on day 21 and day 42 (Figure 1). A more detailed analysis of compliance via urine and other markers is under preparation (Clinicaltrials.gov, no. NCT03384147).

Overall, side effects were minimal. Three participants dropped out; two had a stressful lifestyle due to life-changing events and/or regular night shifts, and one felt uncomfortable after alcohol consumption. Only the latter was related to the alcohol intervention per se. No additional adverse events were reported.

### 3.3. Whole Plasma Lipids

In general, the lipoproteins main and subfractions were highly correlated (Appendix A. No significant effects were observed on total plasma levels of triglycerides, total cholesterol, free cholesterol, cholesteryl esters, or phospholipid (Appendix A).

### 3.4. Lipoproteins and Lipoprotein Subfractions

#### 3.4.1. Apolipoprotein A1 and High-Density Lipoprotein

As shown in Table 3, total plasma apoA1 levels increased following alcohol consumption compared with abstention, *p* = 0.004 (Figure 3). ApoA1 in high-density lipoprotein (HDL), HDL_3_, and HDL_2a_ was also increased (Appendix A). Moderate alcohol intake increased the cholesterol concentration of HDL_3_ and HDL_2a_, *p* < 0.008 (Figure 4 and Figure 5) and the phospholipid content in HDL_2a_ (Appendix A), but HDL_2b_-C and HDL_2b_-apoA1 were unchanged (Appendix A). There was a non-significant 3.9 mg/dL increase in overall HDL-C levels, *p* = 0.19 (Table 3, Appendix A), and the apoB/apoA1 ratio decreased after moderate drinking compared with abstention, *p* = 0.02 (Table 3). Absolute lipoprotein and apolipoprotein values before and after each intervention in periods 1–2 is available in Appendix A.

#### 3.4.2. Apolipoprotein B and Apolipoprotein B-Containing Lipoproteins

The NMR measurements were not reliable for predicting the concentration of individual apoB-containing lipoprotein subfractions, including triglyceride concentration (-TG) in LDL_2-5_, and the phospholipid content of LDL_4_, due to insufficient performance of partial least squares calibration models [36]. No effects on LDL-C, LDL-C subfractions or total plasma apoB levels were found (Appendix A). However, LDL-TG and LDL_1_-TG tended to decrease, *p* = 0.070 and *p* = 0.076. The IDLs, VLDLs, and the particle numbers of apoB-containing lipoproteins were largely unchanged (Appendix A).

#### 3.4.3. Sensitivity Analyses

Adjusting for sex, gram of alcohol per kg body weight or inclusion of a period-subject interaction as random effect did not change the *p*-values, but confounding test showed a sex-difference in analyses for HDL-C; HDL_2a_-C; total apoA; HDL-apoA1; HDL_2b_-apoA1: apoB/apoA1 ratio; and the phospholipid content of HDL_2a_ (*p* < 0.05). Results in drinking versus abstaining women showed similar directions as the overall results, but the *p*-values were attenuated for all HDL-related outcomes (Appendix A). By contrast, the apoB/apoA1 ratio was more robustly decreased compared with the analyses for both sexes (*p* = 0.01), driven by the change in apoA1 content (*p* = 0.06) and a lower number of LDL particles, LDL-P (*p* = 0.04) in drinking versus abstaining women (Appendix A). No significant changes in any LDL-P subfractions or LDL-C measures were found in the analyses of women only (*p* > 0.05). Carry-over was observed for one of the reported variables, HDL_2a_-apoA1 (*p* < 0.05), so the effect on this marker should be evaluated only after period 1; this did not affect the conclusions since HDL_2a_-apoA1 differed between treatments already after period 1.

## 4. Discussion

We analyzed outcomes of the MetAl intervention trial related to lipoproteins in healthy adults after 3 weeks of moderate alcohol consumption compared with abstention in a cross-over design. Similar to other studies [9], a trend towards an increasing effect on HDL-C was found but did not reach significance. However, total plasma apoA1 and apoA1 in HDL, HDL_2a_, and HDL_3_ were significantly higher in drinkers than abstainers. This confirms overall compliance and suggests apoA1 as a more sensitive biomarker than HDL-C of low doses of alcohol intake over a short duration. We also found increased cholesterol content of HDL_2a_ and HDL_3_. Among potentially novel findings were higher phospholipid levels in HDL_2a_, a decreased apoB/apoA1 ratio, and a trend towards a lower LDL-TG and LDL_1_-TG content. Analyses in women alone replicated these results, albeit the effects were weaker. The decrease in the apoB/apoA1 ratio was more pronounced in women alone, explained by a decrease in LDL particle number.

Increased HDL-C levels after an alcohol intake above 12 g/day have been found in several intervention studies with healthy adults [9,13]. A recent meta-analysis of short-term intervention studies and largely healthy individuals found a dose–response relationship and a mean HDL-C difference of 2.8 (0.9–4.6) mg/dL for an alcohol intake between 12.5–29.9 g/day [9]. Likewise, the longest alcohol intervention study showed a 2.0 (1.6–2.2) mg/dL HDL-C increase after 150 ml/d red wine compared with abstention for two years in participants with well-controlled type 2 diabetes [10]. These results compare well with the 3.9 (−1.9–9.8) mg/dL difference in HDL-C we observed after ~12–24 g/day. Still, it seems likely that within only three weeks, the inter-individual variation in response was too large for this difference to become significant (Table 3).

In accordance with the results of MetAl, Schäfer et al. [44] found positive associations between moderate alcohol intake and HDL_2b_-C, HDL_2a_-C, and HDL_3_-C, with the strongest association to HDL_2a_-C, when comparing alcohol doses of 5.1–30 g/day to 0–5 g/day. In a recent systematic review, all HDL-C subfractions were increased after alcohol doses up to 60/d. However, more observational studies reported positive associations for HDL_3_-C than HDL_2_-C [28], suggesting a stronger change in the smallest HDL subfractions.

Alcohol has been suggested to raise HDL-C and the cholesterol carried by its subfractions via mechanisms like higher lipoprotein lipase activity [45], increased plasma apoA1 transport [46], reduced cholesteryl ester transfer protein (CETP) activity [47], or increased cholesterol efflux capacity (CEC) [48]. Increased CEC was suggested as one of the main mechanistic changes following alcohol intakes up to 60 g/day [28]. Whether a stimulated CEC increases HDL-C is unknown. Epidemiological studies show a correlation between HDL-C and CEC of ~0.4 in populations of healthy subjects and CAD patients [49,50], indicating that HDL-C is not a robust measure of CEC. The HDL-C increase could result from an interaction between increased apoA1 and CEC since apoA1 stimulates CEC [51] and is highly correlated with HDL-C [52]. Total apoA1 and apoA1 in HDL and HDL subspecies were increased in MetAl. The alcohol-induced apoA1 is related to increased ethanol-induced hepatic apoA1 production [53,54] and has been ascribed to an upregulated *APOA1* gene expression at the transcription level mediated by acetate or acetyl-CoA [55].

Meta-analyses of intervention studies show increased plasma apoA1 levels following alcohol intake after various doses of alcohol [9,10,13]. However, MetAl is one of the first studies showing significantly increased apoA1 levels with an alcohol dose of only ~12 g/day and a relatively short duration [9]. Other studies have found apoA1 levels increased following alcohol doses of 20–30 g/day after 10–14 days in before-and-after analyses [47,56], or following low doses of alcohol (<15 g/day) with longer duration studies, such as studies lasting 28 days [57] and 2 years [10].

The cardiovascular relevance of increased HDL subfractions and apoA1 remains unresolved. HDL-C is inversely associated with but not causally related to CHD risk [58]. HDL_3_-C, rather than HDL_2_-C, may be responsible for the inverse association between HDL-C and incident CHD, as shown in analyses of cohort studies, although attenuated when adjusting for apoB [59]. HDL_3_-C has also been inversely, and HDL_2_-C directly, related to carotid plaques [60]. However, a review found no improved risk prediction for HDL_2_-C or HDL_3_-C over HDL-C [17]. Interestingly, HDL particles could influence atherosclerosis via mechanisms unrelated to the cholesterol cargo [19,61]. A cross-sectional study found phospholipid enrichment associated with alcohol intake in all HDL fractions, including HDL_2a_ [44]. HDL phospholipid concentration may increase the ability of HDL to promote CEC [62], and lipidomics analyses have a beneficial role in plaque stability and resilience to rupture [63].

ApoA1 is also inversely associated with cardiovascular risk [64]. However, apoA1 could be more relevant in combination with apoB as a cardiovascular risk marker, reflecting the relationship between atherogenic and anti-atherogenic lipoproteins [65]. The apoB to apoA1 ratio has been suggested as superior to traditional risk markers in a case–control study of acute myocardial infarction comprising >20.000 participants [66]. In MetAl, moderate drinking decreased the apoB/apoA1 ratio overall and LDL-P in women. Given the inverse association between apoA1 [64] and HDL-C [58] and the causal relationship between LDL-P and CVD risk [67], such lipoprotein changes might indicate a cardiovascular benefit of moderate alcohol intake. However, other functions of lipoprotein constituents, including apolipoprotein C3 and anti-inflammatory components, might be equally or even more important for HDL and lipoprotein function and should be further investigated [68,69].

A decreased triglyceride content of LDL particles could also reflect a health advantage. Under high triglyceride conditions such as insulin resistance, triglyceride-enriched VLDL stimulates CETP and lipases and the consequent transfer of triglycerides from VLDL to LDL, thereby raising LDL-TG. Subsequently, more small atherogenic LDL particles are formed [70]. High LDL-TG could therefore reflect insulin-resistant states. Higher LDL-TG levels are associated with low-grade systemic inflammation [71] and increased risk of atherosclerotic CVD events in people with and without CVD, pre-diabetes and diabetes [72,73,74]. Conversely, LDL-TG failed to independently predict CVD in secondary analyses of an RCT [75]. However, alcohol doses of 22–24 g/day (~2 units) is associated with a lower risk of type 2 diabetes [76], a condition often characterized by elevated triglycerides [77], and alcohol in doses up to ~40 g/day over periods of 2–12 weeks was found to improve glycemic markers [78]. Thus, a trend towards decreased LDL-TG supports that 1–2 drinks per day (~12–24 g/day) could improve biomarkers related to insulin resistance and type 2 diabetes.

Our study has several strengths, including the cross-over design and the NMR measured lipoproteins with quantitative and qualitative changes. In addition, we used many different alcoholic beverages during the drinking period to ensure that alcohol per se was studied. Besides, the participants demonstrated high compliance with the alcohol intervention and abstention.

A small sample size, a short duration and a non-randomized design are among our limitations, increasing the risk of confounding and allocation bias. The allocation into groups of occasional and habitual drinkers was selected to improve chances for identifying intake biomarkers for different alcoholic beverages but increases the risk of confounding factors separating occasional drinkers from habitual drinkers. Despite this weakness, a range of potential confounding factors was equally distributed between groups at baseline, apart from obvious sex differences, including height. We found significant confounding by sex in several analyses. Therefore, the analyses were replicated in women only, where similar trends in the results were found.

The MetAl study was designed as a pilot study, and we did not monitor food intakes. It is known that major changes in the intakes of fruit, vegetables or wholegrain can affect plasma cholesterol even after a few weeks [79,80,81]. We asked volunteers not to make major changes in their diets, but we cannot rule out that they made changes anyway, and this might distort our results. Dietary changes in plant food intakes typically affect total cholesterol and LDL-cholesterol without affecting HDL or its ratio with total cholesterol. It is, therefore unlikely that such change could lead to the effects observed here, and changes in diets would most likely cause random effects, which would reduce power.

The energy contents of the alcoholic beverages offered differ, but correction for alcohol contents did not affect conclusions in our sensitivity analyses. While beer intake increases carbohydrate intakes by up to 26 g compared to days with strong liqueur, all volunteers had all types of drinks in equal amounts during the alcohol exposure, and the study was randomized with respect to the order of beverages. These factors are, therefore, less likely to affect the results in the current study.

Another limitation is the open design, which was necessary as blinding alcohol consumption is difficult [82].

The MetAl trial aimed to identify new alcohol biomarkers in blood, urine, and hair [83] and was without a wash-out period due to the aim of exploring acute changes in intake biomarkers with changes in alcohol intake. Therefore, the study is prone to carry-over effects, and carry-over was observed for one variable, HDL_2a_-apoA1. Like several other significant variables in our analysis, HDL_2a_-apoA1 was also significant after period 1, and the carry-over effect is, therefore, without importance for our conclusions.

## 5. Conclusions

In summary, moderate alcohol intake raised plasma apoA1, apoA1 content of HDL subfractions, and HDL-C subfractions in healthy adults. ApoA1 seems superior to HDL-C as a compliance marker for short-term low alcohol intakes. Other cardiovascular biomarkers were also changed, including increased HDL_2a_ phospholipid content and a decreased atherogenic to anti-atherogenic lipoprotein ratio. The LDL particle number decreased in women. These findings might indicate that moderate alcohol intake may affect lipoprotein metabolism via changes in lipoproteins related to cholesterol efflux pathways and a minor extent, through cholesterol influx pathways. The results need confirmation in future longer-term, randomized studies exploring sex-stratified effects.

## Figures and Tables

**Figure 1 nutrients-14-05043-f001:**
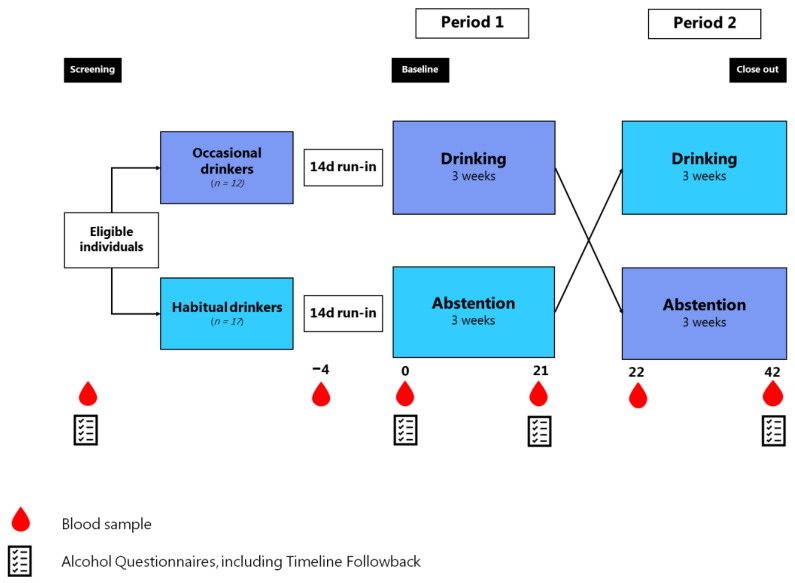
Overview of the MetAl1 study design. Occasional drinkers: habitual alcohol intake <2 drinks (~24 g) per week. Habitual drinkers: habitual alcohol intake ≥2 drinks (~24 g) per week, max 1 drink per day for women and 2 drinks per day for men.

**Figure 2 nutrients-14-05043-f002:**
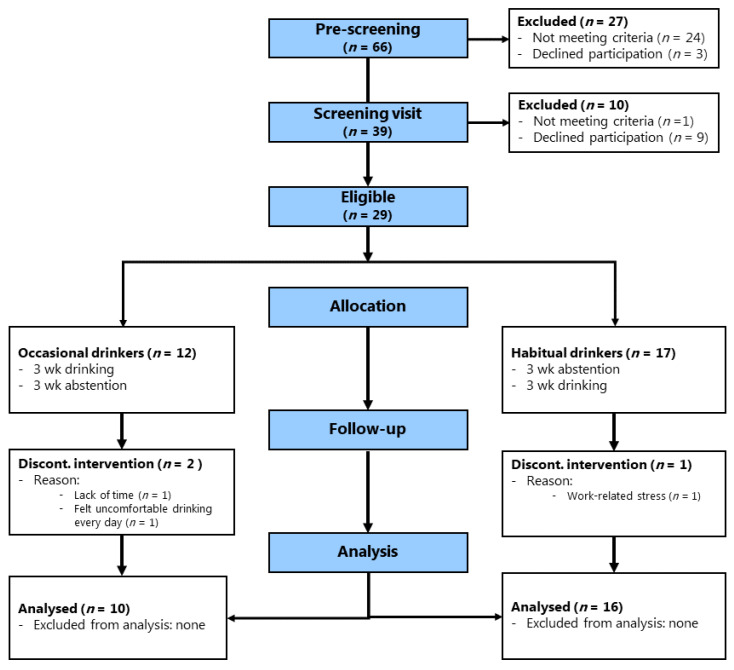
Flow of study participants in MetAl. Occasional drinkers: habitual alcohol intake <2 drinks (~24 g) per week. Habitual drinkers: habitual alcohol intake ≥2 drinks (~24 g) per week, max 1 drink per day for women and 2 drinks per day for men.

**Figure 3 nutrients-14-05043-f003:**
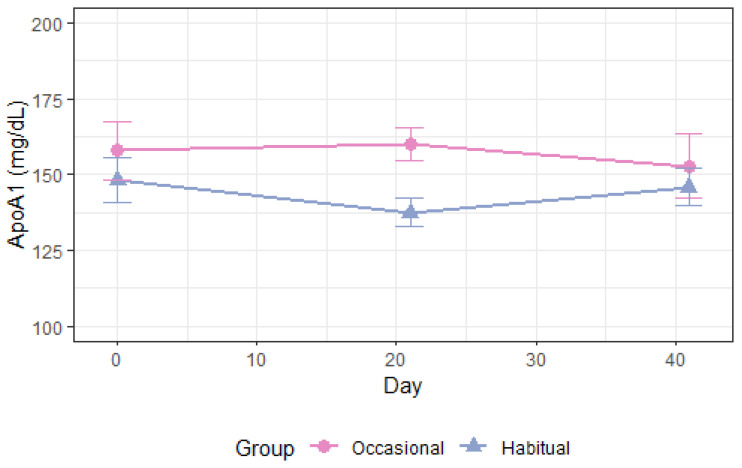
Total apoA1 during 3 weeks in period 1 and 3 weeks in period 2. Occasional drinkers (*n* = 10): habitual alcohol intake <2 drinks (~24 g) per week. Habitual drinkers (*n* = 16): habitual alcohol intake ≥2 drinks (~24 g) per week, max 1 drink per day in women and 2 drinks per day in men. Apo: apolipoprotein.

**Figure 4 nutrients-14-05043-f004:**
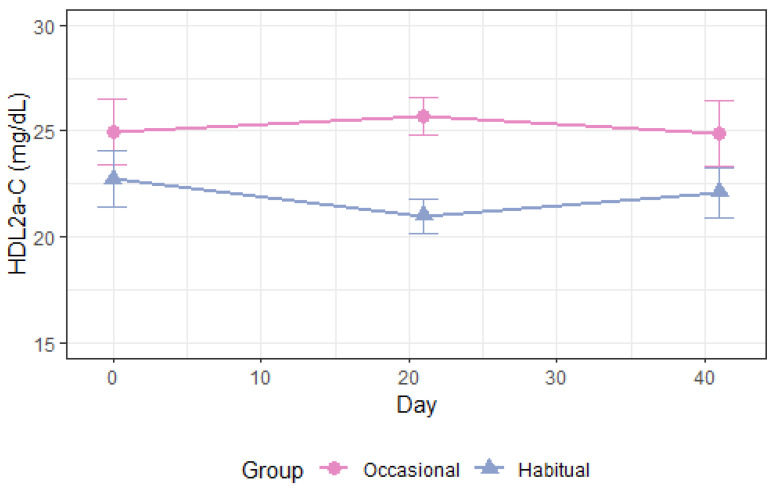
Circulating HDL_2a_-C during 3 weeks in period 1 and 3 weeks in period 2. Occasional drinkers (*n* = 10): habitual alcohol intake <2 drinks (~24 g) per week. Habitual drinkers (*n* = 16): habitual alcohol intake ≥2 drinks (~24 g) per week, max 1 drink per day in women and 2 drinks per day in men. -C: cholesterol concentration, HDL: high-density lipoprotein.

**Figure 5 nutrients-14-05043-f005:**
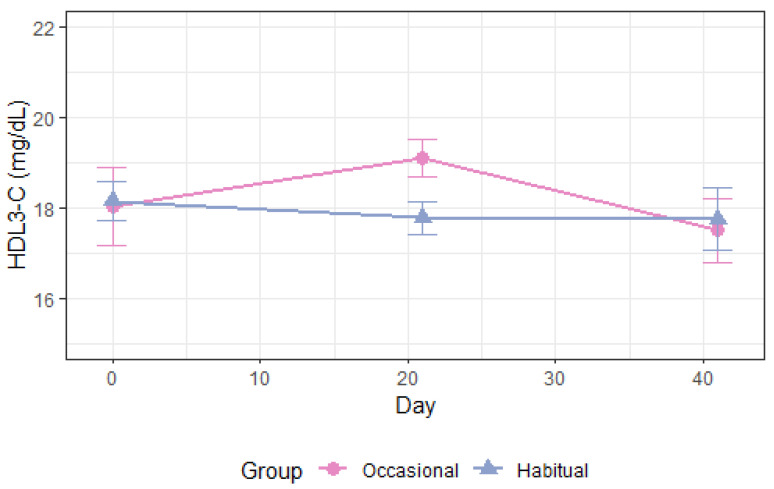
Circulating HDL_3_-C during 3 weeks in period 1 and 3 weeks in period 2. Occasional drinkers (*n* = 10): habitual alcohol intake <2 drinks (~24 g) per week. Habitual drinkers (*n* = 16): habitual alcohol intake ≥2 drinks (~24 g) per week, max 1 drink per day in women and 2 drinks per day in men. -C: cholesterol concentration, HDL: high-density lipoprotein.

**Table 1 nutrients-14-05043-t001:** Standard volumes of different types of alcoholic beverages corresponding to one drink.

Type	Alcohol Volume %	Volume of One Drink (mL)		Alcohol Content (g) ^Δ^
Pilsner beer	4.6	330		12
Strong beer	7.2–8	250		14.2–15.8
Cider	4.6	330		12
Red wine	12–15	130		12.3–15.4
White wine	12–15	130		12.3–15.4
Dessert wine	17–22	80		10.7–13.9
Gin	37–40	40		11.6–12.6
Whiskey	40–46	40		12.6–14.5
Snaps	40–46	40		12.6–14.5

**^Δ^** Estimation based on the density of alcohol of 0.789 g/cm^3^.

**Table 2 nutrients-14-05043-t002:** Baseline characteristics of MetAl participants by group.

Demographic Characteristics	A Occasional Drinkers (*n* = 10)	B Habitual Drinkers (*n* = 16)	*p* B vs. A	C Habitual Drinkers—Women (*n* = 9)	*p* C vs. A
Age, median years (IQR)	30 (22–59)	32 (23–64)	0.38	29 (23–62)	0.90
Female, *n* (%)	10 (100)	9 (56)	0.02	9 (100)	NA
Alcohol intake 14 days run-in, mean drinks * (SD)	5.4 ± 7.3	12.7 ± 8.6	0.03	10.8 ± 7.2	0.12
Smoking status, *n* (%)					
- Never smoked	6 (60)	9 (56)	0.86	5 (56)	0.84
- Former smoker	4 (40)	6 (38)	0.90	3 (33)	0.76
- Current smoker	0 (0)	1 (6)	0.42	1 (11)	0.28
Weight, mean kg (SD)	71 (±17)	69 (±15)	0.76	58 (±7)	0.05
Height, mean cm (SD)	166 (±6)	174 (±9)	0.01	168 (±5)	0.36
BMI, mean kg/m^2^ (SD)	26 (±5)	22 (±3)	0.10	21 (±2)	0.01
Waist circumference, mean cm (SD)	97 (±31)	83 (±10)	0.21	77 (±5)	0.08
HDL-C, mean mg/dL (SD)	62 (±15)	57 (±20)	0.53	63 (±20)	0.86
LDL-C, mean mg/dL (SD)	91 (±18)	90 (±22)	0.92	76 (±16)	0.08
TG, mean mg/dL (SD)	100 (±33)	105 (±46)	0.76	86 (±26)	0.33
TC, mean mg/dL (SD)	183 (±28)	179 (±33)	0.73	165 (±33)	0.24
Systolic BP, mean mmHg (SD)	108 (±9)	117 (±12)	0.07	112 (±10)	0.44
Diastolic BP, mean mmHg (SD)	70 (±6)	72 (±8)	0.68	71 (±8)	0.88
Pulse, mean bpm (SD)	75 (±10)	67 (±9)	0.07	70 (±9)	0.29

* ~12 grams of alcohol per drink. Occasional drinkers (*n* = 10): habitual alcohol intake <2 drinks (~24 g) per week. Habitual drinkers (*n* = 16): habitual alcohol intake ≥2 drinks (~24 g) per week, max 1 drink per day in women and 2 drinks per day in men. BMI: body mass index, BP: blood pressure, -C: cholesterol concentration, HDL: high-density lipoprotein, IQR: interquartile range, LDL: low-density lipoprotein cholesterol, NA: not applicable, SD: standard deviation, TC: total cholesterol, TG: triglyceride.

**Table 3 nutrients-14-05043-t003:** Effects of 1–2 drinks (~12–24 g) per day on NMR measured HDLs, apoA1, and HDL subfractions before and after alcohol intake or abstention and in drinking periods compared with abstention.

	Period 1		Period 2		Period 1 + 2	
	ΔMean1 ^£^	*p*	ΔMean2 ^¤^	*p*	ΔMean Drinking vs. Abstention *	*p*
	Mean (95% CI)		Mean (95% CI)		Mean (95% CI)	
HDL-C, mg/dL Occ. vs. Hab. All	−4.8 (−13.1–3.5)	0.26	−3.1 (−11.3–5.2)	0.46	−3.9 (−9.8–1.9)	0.19
HDL_2b_-C, mg/dL Occ. vs. Hab. All	−1.6 (−7.2–4.1)	0.59	2.3 (−3.4–7.9)	0.43	0.4 (−3.6–4.4)	0.86
HDL_2a_-C, mg/dL Occ. vs. Hab. All	−2.7 (−5.0–[−0.5])	0.02	−1.9 (−4.1–0.3)	0.09	−2.3 (−3.9–[−0.7])	0.004
HDL_3_-C, mg/dL Occ. vs. Hab. All	−1.6 (−3.2–0.09)	0.06	−1.6 (−3.2–0.04)	0.056	−1.6 (−2.7–[−0.4])	0.008
HDL_2b_-TG, mg/dL Occ. vs. Hab. All	0.2 (−0.6–1.1)	0.62	−0.1 (−1.0–0.7)	0.77	0.04 (−0.6–0.6)	0.89
HDL_2a_-TG, mg/dL Occ. vs. Hab. All	−0.001 (−0.7–0.7)	0.99	−0.8 (−1.5–[−0.05])	0.04	−0.4 (−0.9–0.1)	0.14
HDL-PL, mg/dL Occ. vs. Hab. All	−7.8 (−21.9–6.3)	0.28	−5.4 (−19.4–8.6)	0.45	−6.6 (−16.5–3.3)	0.19
HDL_2b_-PL, mg/dL Occ. vs. Hab. All	−3.5 (−13.8–6.7)	0.50	1.1 (−9.1–11.2)	0.83	−1.2 (−8.4–6.0)	0.74
HDL_2a_-PL, mg/dL Occ. vs. Hab. All	−4.5 (−9.7–0.78)	0.10	−3.3 (−8.5–1.9)	0.22	−3.9 (−7.6–[−0.2])	0.04
ApoA1, mg/dL Occ. vs. Hab. All	−13.8 (−27.9–0.3)	0.055	−15.5 (−29.5–[−1.5])	0.03	−14.7 (−24.6–[−4.7])	0.004
HDL-apoA1, mg/dL Occ. vs. Hab. All	−12.1 (−24.7–0.4)	0.058	−12.1 (−24.5–0.4)	0.057	−12.1 (−21.0–[−3.3])	0.007
HDL_2b_-apoA1, mg/dL Occ. vs. Hab. All	−3.4 (−12.0–5.1)	0.43	1.2 (−7.2–9.7)	0.78	−1.1 (−7.1–4.9)	0.72
HDL_2a_-apoA1, mg/dL Occ. vs. Hab. All	−5.6 (−10.2–[−1.0])	0.02 ^#^	−5.3 (−9.8–[−0.7])	0.02	−5.4 (−8.7–[−2.2])	0.001 ^#^
HDL_3_-apoA1, mg/dL Occ. vs. Hab. All	−3.2 (−7.9–1.5)	0.18	−7.8 (−12.4–[−3.1])	0.001	−5.5 (−8.8–[−2.2])	0.001
ApoB/apoA1, mg/dL Occ. vs. Hab. All	0.11 (0.03–0.19)	0.01	0.08 (−0.005–0.16)	0.07	0.09 (0.03–0.15)	0.002

Occasional drinkers (*n* = 10): habitual alcohol intake <2 drinks (~24 g) per week. Habitual drinkers (*n* = 16): habitual alcohol intake ≥2 drinks (~24 g) per week, max 1 drink per day in women and 2 drinks per day in men. ^£^ ΔMean1: mean change in occasional drinkers compared with mean change in habitual drinkers in period 1. ^¤^ ΔMean2: mean change in occasional drinkers compared with mean change in habitual drinkers in period 2. * Mean changes in lipids and lipoproteins in drinking participants compared with abstaining participants in both periods, independent of group relationship. ^£,¤,^* Values for drinking periods were subtracted from abstaining, and negative numbers therefore indicate increased levels while drinking compared with abstaining. ^#^ Carry-over was observed for this variable so the effect should be evaluated after period 1. Apo: apolipoprotein, -C: cholesterol concentration, Hab: habitual, HDL: high-density lipoprotein, Occ: occasional, -PL: phospholipid concentration, -TG: triglyceride concentration.

## Data Availability

The data generated and analyzed for this study are can be accessed upon request to the corresponding author.

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
