# Peer review of "1–2 Drinks Per Day Affect Lipoprotein Composition after 3 Weeks—Results from a Cross-Over Pilot Intervention Trial in Healthy Adults Using Nuclear Magnetic Resonance-Measured Lipoproteins and Apolipoproteins"

_nutrients, 2022, doi:10.3390/nu14235043_

Round 1

Reviewer 1 Report

1. Although the presented study is interesting, it has quite big limitations. The first is the small size of the group, and the second is the short duration of the study. I believe that this should be emphasized and reflected both in the title, adding the information that it was only a pilot study, and throughout the work. Especially in the discussion of the results, the above-mentioned limitations should be clearly emphasized.

2. The sequence of tables in the text should start with 1. It should be corrected.

3. There is no explanation of NA abbreviation in Table 1.

4. The quality of Fig. 2 should be improved and explanations of all abbreviations added.

5. All abbreviations used in the text should be explained in the Abbreviations section.

Author Response

We thank the reviewers for their careful work and we hope now to have addressed their major concerns.

Reviewer 1.

  1. Although the presented study is interesting, it has quite big limitations. The first is the small size of the group, and the second is the short duration of the study. I believe that this should be emphasized and reflected both in the title, adding the information that it was only a pilot study, and throughout the work. Especially in the discussion of the results, the above-mentioned limitations should be clearly emphasized.

We have added ’pilot’ in the title and extended the discussion of the limitations of this study.

  1. The sequence of tables in the text should start with 1. It should be corrected.

This flaw has now been corrected.

  1. There is no explanation of NA abbreviation in Table 1.

This has now been added.

  1. The quality of Fig. 2 should be improved and explanations of all abbreviations added.

Figure 2 has now been enlarged to make the text more readable.

  1. All abbreviations used in the text should be explained in the Abbreviations section.

Thanks for spotting! All abbreviations should now b elisted also in the Abbreviations list. Some abbreviations that were not used more than once have been omitted.

Reviewer 2 Report

The present study is a cross-over trial of moderate alcohol consumption vs. abstinence for a period of 3 weeks and its effects on lipoprotein levels in 26 healthy adults.

The results showed that moderate alcohol consumption increased HDL2a-C, HDL3-C,  total apoA1 and apoA1 content of HDL subfractions and HDL-C. Also, an increase in HDL2a and a decrease in ApoB/apoA1 ratio was observed.

Major points

The research question is relevant, but there are some serious concerns regarding the study design. The small sample size, the short duration of the study and the absence of wash-out-period raise important concerns regarding the importance of these results. Due to the latter, the chances of a carry over effect are substantial.

Minor points

Introduction:                              

Rows 45-47: here you contradict yourself. The first and second sentenced say the same thing, but are linked with a “however”. Please reformulate

Materials and Methods:

Row 90: What do you mean with practical reasons of changing the protocol of the run-in period from 4 to 14 days?

Did the participants lose weight? Were there any changes in their dietary habits during the intervention period? These could be seen as potential confounders, even if they were advised not to change their habits.

Did you take into consideration that different kinds of alcoholic beverages provide different energetic content? This could also a source of confounding.

Please discuss further on why a wash-out period was not planned.

Tables 1 and 2 are presented in reverse order, should be corrected.

Author Response

We thank the reviewers for their careful work and we hope now to have addressed their major concerns.

Reviewer 2

Major points

The research question is relevant, but there are some serious concerns regarding the study design. The small sample size, the short duration of the study and the absence of wash-out-period raise important concerns regarding the importance of these results. Due to the latter, the chances of a carry over effect are substantial.

We agree and carry-over has now been tested; despite the large number of tests only one variable was showing carry-over, and this variable was also significant after period 1, so the effect was quite robust. We share the concerns of the reviewer about the study size, but in fact many of the findings in this s tudy that we conclude on are also significantly different already after period 1. So if there was carry-over that could not be identified by the standard test used, we would most likely still have the same conclusion. A more extensive discussion of carry-over and its implications has been added.

Minor points

 Introduction:                             

Rows 45-47: here you contradict yourself. The first and second sentenced say the same thing, but are linked with a “however”. Please reformulate

We have now reformulated this; the point is that despite beneficial effects of alcohol on lipoproteins this may not translate into disease prevention or explain the J-shaped curve characterizing alcohol intake and CVD risk. The sentence is meant to avoid overstating the importance of lipoproteins in explaining the effects of alcohol since this is a very sensitive issue. The dilemmas in explaining this relationship are outlined in the rest of the paragraph.

Materials and Methods:

Row 90: What do you mean with practical reasons of changing the protocol of the run-in period from 4 to 14 days?

We have alcohol intake records for the 14 days between screeing and baseline, and for the current study it gives more sense to view baseline as the average intakes over two weeks, because this period is more comparable to the intervention periods than just 4 days.

Did the participants lose weight? Were there any changes in their dietary habits during the intervention period? These could be seen as potential confounders, even if they were advised not to change their habits.

We did not measure weights at the end of the study since we did not think weight changes (or lack of weight changes) over just 6 weeks would be interpretable.

Did you take into consideration that different kinds of alcoholic beverages provide different energetic content? This could also a source of confounding.

Yes, this was taken into consideration by providing all alcoholic beverages to all participants in equal amounts, but in random orders. So, they all received the same energy through their alcohol intervention period. This has now also been added in the discussion, since it is an important point.

Please discuss further on why a wash-out period was not planned.

The study was planned to look at very short-term (1-2 days) as well as slightly longer-term (3 weeks) effects om a range of biomarkers. By avoiding wash-out, we can look at short-term changes for relevant biomarkers such as alcohol intake biomarkers or possibly hormones; but changes in lipoproteins over a single day is of course not relevant.

Tables 1 and 2 are presented in reverse order, should be corrected.

Yes, this flaw has now been corrected, thanks.

Round 2

Reviewer 1 Report

I accept in present form.